# Effect of TiO_2_ on the Sintering Behavior of Low-Grade Vanadiferous Titanomagnetite Ore

**DOI:** 10.3390/ma14164376

**Published:** 2021-08-05

**Authors:** Songtao Yang, Weidong Tang, Xiangxin Xue

**Affiliations:** 1School of Materials and Metallurgy, University of Science and Technology Liaoning, Anshan 114051, China; 2School of Metallurgy, Northeastern University, Shenyang 110819, China; ning6506@163.com

**Keywords:** low-grade vanadiferous titanomagnetite ore, sinter, perovskite, mineral phase

## Abstract

Low-grade vanadiferous titanomagnetite ore (LVTM) is as an important mineral resource for sintering ore manufacturing. Furthermore, TiO_2_ has a significant effect on the sintering process of iron ore fines. The effects of TiO_2_ on the metallurgical properties, microstructure, and mineral composition of LVTM sinter were investigated by sintering pot tests, X-ray diffraction (XRD), scanning electron microscopy (SEM), and mineral phase microanalysis. The results were as follows: as the TiO_2_ content increased from 1.75% to 4.55%, the flame front speed and productivity decreased, while the reduction degradation index (RDI) and softening properties deteriorated. In addition, the tumbler index (TI) values reached a maximum at TiO_2_ = 1.75%. In addition, with increasing TiO_2_ content, an increase in the magnetite and perovskite phase, and a decrease in calcium ferrite and hematite were found with an increase in TiO_2_ content. Thus, the lower the TiO_2_ content, the better the quality of the sinter.

## 1. Introduction

Vanadiferous titanomagnetite ore (VTM) provides an important source of iron, and the associated vanadium, titanium, chromium, cobalt, nickel, platinum, and scandium components are of high economic value. The VTM is distributed globally with large reserves, mainly in China’s Panzhihua and Chengde regions, exceeding five billion tons [1,2,3,4,5]. However, after years of mining, Chengde’s main mining areas have been emptied of high-grade VTM and there is little prospect of finding ore in the periphery of the mine. Recently, with the continuous advancement of mining technology and soaring mineral resource prices, the value of the large amount of low-grade vanadiferous titanomagnetite ore (LVTM) identified within the mine production areas has been highlighted by steel companies. The LVTM resources in the Chengde area have been estimated to exceed 10 billion tons [6,7,8,9]. Multiple studies have been done on conventional VTM, while fewer studies on LVTM with complex phase compositions and different properties are available [5,6]. Therefore, it is crucial to study the utilization of LVTM to reduce the production costs of steel enterprises and for the sustainable development of China’s steel industry.

To use LVTM efficiently, the authors’ laboratory proposed the sintering behaviors of LVTM [7,8,9,10]. In the blast furnace (BF) process where iron ore is reduced to molten iron, the LVTM sinter with different basicity, coke ratios, and MgO contents have been studied. Yang [8,9] studied the effects of coke ratios and MgO on the sintering behaviors and the metallurgical properties of LVTM. Yang [10] surveyed the effect of basicity on the mineral compositions and element migration of LVTM sinters. Particularly, titanium is one of the main elements in LVTM, influencing its sintering properties. The perovskite was generally cubic or octahedral, with an ideal cubic microstructure and a slightly twisted octahedron. The titanium ions were at the body center of the cubic cell, the oxygen ions was at the face center, and the calcium ions were at the top of the corners. The perovskite cubic crystals often had parallel crystal edges, which was the result of flake bicrystals at the process of the high-temperature variant changes to the low-temperature variant [11,12]. Perovskite has a high melting point (2243 K) and is the first to precipitate during the cooling process, and it does not have a bonding effect itself, however, it acts as a sort of “crystalline” interface between magnetite and hematite particles. This “continuous crystal” interface is easily destroyed. Controlling and limiting the formation of calcium ferrite is important to improve the quality of the vanadiferous titanomagnetite ore sinter. However, the TiO_2_ contents in these studies were lower than 2 wt%, and the sintering behaviors with a relatively higher TiO_2_ content have not been studied comprehensively. Therefore, it is necessary to investigate the sintering behavior of LVTMs with high titanium dioxide content.

As part of continuing work to study the utilization of LVTM in the BF process, this paper explores the effect of TiO_2_ on the sintering behavior of LVTM. First, the yield (+5 mm), weight loss, flame front speed, productivity, tumbler index (TI), reduction degradation index (RDI), reduction index (RI), and softening properties were examined. The mineral compositions and microstructures of the LVTM sinter with different TiO_2_ contents were studied. Furthermore, the sinter with different TiO_2_ content was evaluated. These results will provide theoretical and technical bases for the effective production of the LVTM sinter and facilitate its exploitation.

## 2. Materials and Methods

### 2.1. Materials

The raw materials used in the sintering pot test including LVTM, high titanium iron ore powder (HT ore), ordinary iron ore (Indian ore, Malaysian ore, Zirong ore), dolomite, gas ash, vanadium extraction tailings (V tailings), quicklime, and coke were purchased from the Jianlong Iron and Steel Co. Ltd. (Chengde, China). Table 1 shows the chemical composition of the raw materials used for the sintering experiments. Table 2 shows the proximate analysis of the coke breeze and the ash composition. The total iron content (TFe) of LVTM was 63.50%, the Cr_2_O_3_ content of LVTM was 0.08%, and the TiO_2_ content of LVTM was 2.18% lower than that of Panzhihua VTM [8,9]. The Cr_2_O_3_ content of the LVTM was 0.08%. The TiO_2_ content of the blended material was adjusted by preparing high titanium (HT) ore with a 6.22% TiO_2_ content.

### 2.2. Sintering Pot Test

Figure 1 shows the sintering pot experimental apparatus diagram, which performs the sintering operation using air extraction and negative pressure. The detailed sintering parameters for the sintering pot test and the sintering experimental programs are shown in Table 3 and Table 4, respectively. The raw material ratios were determined from production data provided by Jianlong Steel Co(Chengde China). The quicklime basicity (*w* (CaO)/*w* (SiO2)) was adjusted to 1.9, the carbon content was 3.2%, and the return fines added were 18%. The mass percentage content of Indian ore, Malaysian ore, Zirong ore, return fine, ash, V tailings, and dolomite for all experimental groups was 5%, 4%, 5%, 18%, 1%, 2%, and 1%, respectively, while the mass percentage content of LVTM ranged from 0% to 64% and the mass percentage content of HT ore ranged from 0% to 60%. The TiO_2_ content of the experimental feedstock was adjusted by adjusting the mass percentage content of LVTM and HT ore. The effect of different TiO_2_ contents (1.75–4.55%) on the sintering process and metallurgical properties of LVTM was investigated in experimental steps of 0.7%. The sintering pot test steps are, in order, burdening, mixing, granulation, ignition, sintering, cooling, and crushing [2]. The raw material mixture (100 kg) was placed in a sintering pot, ignited with natural gas to 1373 K, and held at a negative pressure of 6 kPa for 120 s. Sintering was carried out at a negative pressure of 10 kPa and ended when the sinter tail gas temperature reached a maximum. After sintering, the sinter was air-cooled for 10 min, then poured out of the sintering pot for crushing. The sintering process parameters included flame front speed (V, mm/min), weight loss (L, %), yield (+5 mm) (Y, %), and productivity (P, t/(m^2^·h). The flame front speed reflects the speed of sintering time. The weight loss reflects the mass loss of ore during sintering. The yield (+5 mm) reflects the number of finished sintered ores with a passable particle size of >5 mm. The productivity of sinter is an important indicator to measure the production capacity of sintering production. The flame front speed (V, mm/min) was calculated by Equation (1):(1)$$V=\frac{h_{1}-h_{2}}{t}$$
where h_1_ is the total height of the sintered pot (mm); h_2_ is the height of the material-free cup after filling (mm); and t is the sintering test time (min). The weight loss (L, %) was calculated by Equation (2):(2)$$L=\frac{m_{1}-(m_{2}+m_{3})}{m_{1}}\times 100\%$$
where m_1_ is the mass of the raw material (kg); m_2_ is the mass of the sinter (kg); and m_3_ is the mass of the bedding material (kg). In this test, m_3_ was 4.0 kg. The yield (+5 mm) (Y, %) was calculated by Equation (3):(3)$$Y=\frac{W_{1}}{m_{2}}$$
where W_1_ is the mass of the sinter with a particle size greater than 5 mm (kg). The productivity (P, t/(m2 h) was calculated by Equation (4):(4)$$P=\frac{m_{2}\times Y}{S\times t}$$
where S is the cross-section area of the sintering pot (m^2^).

The sinter cake was crushed using a jaw crusher, and then the material was crushed and tested by dropping it three times from 2000 mm. Finally, the sinter material was divided by an automatic sieving device into six sizes: >40 mm, 25–40 mm, 16–25 mm, 10–16 mm, 5–10 mm, and <5 mm.

### 2.3. Metallurgical Properties Test and Mineral Phase Analysis

The sinter were tested for tumble and abrasion (TI), reduction-disintegration (RDI), and reduction (RI) indexes, according to ISO-3271, ISO-4696 and ISO-7215, respectively. The TI of sinter reflects the ability of the sinter to resist extrusion at room temperature. The RDI reflects the BF in the upper stack regions where it is mildly reducing, and temperatures are low. The particle sizes (+6.3 mm, +3.15 mm, 0.5 mm) of sinter after reduction were used to calculate the RDI and expressed as RDI_+6.3_, RDI_+3.15_, and RDI_−0.5_, respectively. The RI for sinter reflects the reduction of sinter in the blast furnace at 1173 K. The mineral phases of the samples were observed by polarized light microscopy (Cambridge Q500, Leica Microsystems, Germany), X-ray diffraction (XRD) analysis (MPD/PW3040, Panalytical, Netherlands), and scanning electron microscopy (SEM) (S-3400N, JEOL Ltd., Tokyo, Japan). XRD analysis was quantified using Cu Kα radiation (λ = 0.15406 nm) (40 kV, 50 mA, 2θ = 10–90°).

Figure 2 shows a diagram of the softening properties of the experimental apparatus. According to the actual temperature and atmosphere of sinter in the blast furnace and considering the safety of the experimental equipment, the experimental atmosphere and temperature parameters were formulated, as shown in Table 5. The temperature at which the height of the test specimen shrinks by 10% was defined as the softening start temperature (T_10_%). The temperature at which the height of the specimen shrinks by 40% was defined as the softening end temperature (T_40_%). The softening temperature range (ΔT) was T_10_% to T_40_%. The specimen sample consists of a 40 mm high cylinder of sintered ore with a particle size range of 2.5–3.2 mm and a load of 1 kg/cm^2^.

## 3. Results and Discussion

### 3.1. Sintering Process Parameters

The composition of sinter with different TiO_2_ contents was obtained by sinter pot experiments according to the experimental scheme in Table 4, as shown in Table 6. The X-ray fluorescence (XRF, ZSXPrimus II; Rigaku, Tokyo, Japan) was used to test the chemical compositions of raw materials. As the TiO_2_ content increased, the sinter showed a slight decrease in total iron (TFe) and MgO and a slight increase in CaO, SiO_2_, Al_2_O_3_, and V_2_O_5_.

Figure 3 shows the process parameters of the LVTM sintering process at different TiO_2_ contents. With the TiO_2_ content increase, the flame front speed decreased from 16.67 mm/min to 12.00 mm/min and the yield (+5 mm) decreased from 77.02% to 69.79%. The flame front speed decreased mainly due to the increase in viscosity of the sintering liquid phase, the decrease in the permeability of the material layer, and the material resistance increase. In addition, as the flame front speed decreased, the sintering time increased, and the material loss in the sintering process increased slightly, therefore, the weight loss also increased gradually from 9.08% to 10.27%. Combining all these factors, the productivity decreased from 1.27 t/m^2^-h to 1.19 t/m^2^-h as the TiO_2_ content increased.

### 3.2. Size Distribution

Figure 4 shows the particle size distribution of the sinter at different TiO_2_ contents. The TiO_2_ content increased from 1.75% to 4.55%, and the average particle size of the sinter decreased from 16.94 mm to 12.56 mm. As the TiO_2_ content increased, the liquid phase became less mobile, resulting in holes in the sinter after cooling, leading to inferior sintering ore fall properties and smaller particle size material, while the carbon allocation conditions remained unchanged.

### 3.3. Microstructure and Mineralogy

Figure 5 shows the XRD patterns of sinters with different TiO_2_ contents. The main phases of the sinter were magnetite, hematite, perovskite, calcium ferrite, and silicate, with a small amount of ilmenite. The sinter composition varied with TiO_2_ content. Under the current experimental conditions, the sintered mineral composition of the LVTM sinter obtained from the sintering pot experiments was determined microscopically, and the results are shown in Figure 6. As the TiO_2_ content of the sinter increased, magnetite slightly increased, hematite slightly decreased, perovskite increased, and calcium ferrite decreased.

The perovskite can be formed in a variety of ways. Thermodynamic analysis showed that reactions between CaO-TiO_2_, CaO-FeO-TiO_2_, CaO-2FeO-TiO_2_, CaO-Fe_2_O_3_-TiO_2_, CaO-Fe_2_O_3_-FeO-TiO_2_, and CaO-Fe_2_O_3_-2FeO-TiO_2_ all produced perovskite (CaO-TiO_2_) [13,14]. In practice, however, the perovskite formation was determined by the kinetic conditions of the reaction. The above reactions can be divided into solid–solid and liquid–solid phase reactions, where liquid–liquid phase reactions are also possible. Through the solid–solid reaction, perovskite is formed at a slow rate and high temperatures (1593–1693 K), the solid-phase reaction between CaO and TiO_2_ was accelerated, and the sinter showed perovskite formation. It should be noted that other low-melting liquid phases may partially or entirely melt this solid-phase reaction at high temperatures, and the calcium-titanium ore phase precipitates first during cooling, which is different from the solid-phase reaction-generated calcium-titanium ore. In addition, in LVTM concentrates, TiO_2_ mainly exists as FeO-TiO_2_ and 2FeO-TiO_2_, with melting points of 1633 K and 1743 K, respectively. CaO reacts with Fe_2_O_3_ to form low melting point calcium ferrites, so that reactions may occur between the liquid–solid (reactions 11 and 12) and liquid–liquid phases (reactions 10 and 11) to produce stable, high melting point calcium ferrite precipitates from the liquid phase. Under high temperature and reducing atmospheric conditions, calcium ferrite decomposes and reduces, allowing more CaO to react with TiO_2_-containing minerals to form perovskite. Therefore, under constant basicity and carbon content, the TiO_2_ content of the mixture should be reduced, reducing the formation of calcium titanite and improving the performance of the sinter.
2FeO (s) + TiO_2_ (s) = 2FeO·TiO_2_ (s)(5)
FeO (s) + TiO_2_ (s) = FeO·TiO_2_ (s)(6)
CaO (s) + 2FeO·TiO_2_ (s) = CaO·TiO _2_ + 2FeO(7)
CaO (s) + FeO·TiO_2_ (s) = CaO·TiO_2_ (s) + FeO(8)
CaO·Fe_2_O_3_ (s) + TiO_2_ (s) = CaO·TiO_2_ (s) + Fe_2_O_3_(9)
2FeO·TiO_2_ (s) + CaO·Fe_2_O_3_ (s) = CaO·TiO_2_ (s) + Fe_3_O_4_ + FeO(10)
CaO·Fe_2_O_3_ (s) + FeO·TiO_2_ (s) = CaO·TiO_2_ (s) + Fe_2_O_3_ + FeO(11)
CaO·Fe_2_O_3_ (s) + SiO_2_ (s) = CaO·SiO_2_ (s) + Fe_2_O_3_(12)

The crystallization pattern of the LVTM sinter was complex. The microstructure of the sinter was shown to be mostly granular, interwoven fused, and partially wrecked crystal structures (Figure 7), needle-like interwoven structures were rare, and the porosity was high. With the increase in TiO_2_ content, the liquid phase and fused structures decreased and the porosity increased. The microstructure data show that with TiO_2_ content, the iron-bearing minerals were still dominated by a crystal shape. However, the uniformity of distribution in the sintered ore decreased, the perovskite increased significantly, and grains became coarser. The calcium ferrate changed to short plate-like and columnar structures, and the content decreased further.

### 3.4. Metallurgical Performance

Figure 8 shows the metallurgical performance of the sinter at different TiO_2_ contents. As shown in Figure 8a,b, the RDI_+3.15_ index decreased and the RI index increased slightly as the TiO_2_ content increased. The conversion of the hematite present in the titanium-bearing sinter from a tricrystalline hexagonal lattice to magnetite in an equiaxed tetragonal lattice is generally accepted as the main reason for the deterioration of the RDI properties of sintered ores by TiO_2_. Moreover, the lattice transformation causes structural distortion and volume expansion, resulting in significant internal stresses leading to severe rupture under physical action [15,16,17,18]. Second, perovskite is dispersed between the silicate and iron minerals, weakening the bonding effect of silicate and the crystallization effect of hematite and magnetite. Finally, TiO_2_ and Al_2_O_3_ were completely dissolved in the silicate phase, significantly destroying the fracture toughness, and enlarging the crack. Additionally, the higher the TiO_2_ content, the stronger these destructive effects. On the other hand, with the increase in TiO_2_ content, hard-to-reduce magnetite increased, and the easy-to-reduce titanic hematite and calcium ferrite decreased. In contrast, the content of high melting point vein-like mineral phases and calcium titanite increased, so that the RI of sinter decreased. However, with the increase of TiO_2_ content, the liquid phase decreased, the porosity increased, conducive to reduction, so the RI of sintered ore increased slightly.

As shown in Figure 8c, the TI of the sinter decreased with an increase in TiO_2_ content. The liquid phase generated during the melting of the mixture had high liquidity, high content, and low viscosity, which is beneficial to particle bonding, thus improving the strength of the sinter [19,20,21,22]. At a constant fuel level, as the TiO_2_ content increased, the amount of liquid phase produced decreased, the liquid phase became less fluid and less uniformly distributed, the bonding force weakened, and the pore space of sintered ore increased, leading to the deterioration of the performance of sintered ore. In addition, with the increase in TiO_2_ content, the brittle and hard perovskite content increased, leading to the deterioration of sinter drum strength.

Figure 9 shows the softening indexes of the sinter with various TiO_2_ contents. With the increase in TiO_2_ content, T_10%_ and T_40%_ of the sinter rose, and ΔT became broad. When the TiO_2_ content increased from 1.75% to 4.55%, T_10%_ rose by 24 K, T_40%_ rose by 40 K, ΔT increased by 16 K, and the softening performance of the sinter worsened. The softening characteristics of the sinter during the warming reduction process mainly depend on the amount of high and low melting point minerals produced in the process, and the high melting point minerals mainly influence the softening temperature of the sinter. As the content of high melting point minerals including CaO-TiO_2_ and titanium garnet in sinter increased, the softening temperature of the sintered ore also increased.

### 3.5. Comprehensive Index

The TiO_2_ content in the sinter influences its quality. The objective weights of the indicators were determined by the Delphi method, and the objective weights of the indicators were determined by the entropy method. The comprehensive weighting score could accurately obtain the most suitable TiO_2_ content. The Delphi method was a method of obtaining consensus through a survey of experts, and the entropy method was a mathematical method used to determine the degree of dispersion of an indicator; detailed steps can be found in [23,24]. The steel companies determined the weight coefficients of the selected indicators (P, TI, RDI, and RI). Let X be the evaluation matrix, Z be the criterion matrix, α be the subjective weight, β be the objective weight, and W be the total weight. This study requires P, TI, RDI, and RI to be as large as possible, so X was standardized according to the criterion that the larger the overall weighted rating value is better, to obtain the final evaluation matrix Z.

First, the values of the selected indicators were incorporated into the matrix X = (x_ij_),
(13)$$X_{\mathrm{ij}}=\left\{ \begin{array}{llll} 1.27 & 61.17 & 58.83 & 73.85\\ 1.22 & 58.63 & 51.06 & 75.25\\ 1.24 & 55.14 & 41.35 & 76.24\\ 1.23 & 53.59 & 39.19 & 76.98\\ 1.20 & 51.5 & 37.53 & 77.84 \end{array}\right.$$

Second, normalizing X gives Z = (z_ij_),
(14)$$Z_{\mathrm{ij}}=\left\{ \begin{array}{llll} 0.44 & 0.43 & 0.53 & 0.00\\ 0.12 & 0.32 & 0.34 & 0.13\\ 0.25 & 0.16 & 0.09 & 0.22\\ 0.19 & 0.09 & 0.04 & 0.29\\ 0.00 & 0.00 & 0.00 & 0.37 \end{array}\right.$$

The subjective weight α = (α_1_, α_2_, …, α*_m_)* ^T^.

Define:(15)$$\sum_{j=1}^{m}\alpha_{j}=1,\alpha \geq \;0\;(j=1,\;2,\ldots ,m)$$

The significance coefficients of the indicators, P, TI, RDI, and RI were 0.1, 0.3, 0.5, and 0.1, respectively, yielding α = [0.1, 0.3, 0.5, 0.1]^T^.

The objective weights of each indicator were determined by the entropy value method β,
β = [0.21, 0.24, 0.43, 0.19]^T^(16)

Third, a preference coefficient of 0.5 was set in this study to obtain the combined weight of each indicator W,
W = [0.16, 0.27, 0.43, 0.14]^T^(17)

Finally, the comprehensive evaluation f_i_ was calculated according to the following formula:(18)$$f_{i}=\sum_{j=1}^{m}z_{\mathrm{ij}}w_{j}=1,i=1,\;2,\;\ldots ,n$$

The calculation values are shown in Figure 10. As the TiO_2_ content of the sinter gradually increased from 1.75 to 4.55, the comprehensive index was 85.63, 56.71, 35.43, 27.17, and 14.37, respectively. The optimum energy efficiency was obtained at 1.75% TiO_2_ content.

## 4. Conclusions

The results of this study led to the following conclusions:When the LVTM sinter TiO_2_ content increased from 1.75% to 4.55%, the flame front speed, P, Y, TI, RDI, and softening properties decreased and RI improved.The mineral composition of the LVTM sinter changed considerably when the TiO_2_ content was varied. As the TiO_2_ content increased, the magnetite and perovskite phases increased, while the hematite and calcium ferrite phases decreased.As the TiO_2_ content increased, the comprehensive index of the sinter decreased. In this study, the appropriate TiO_2_ content was 1.75%.

## Figures and Tables

**Figure 1 materials-14-04376-f001:**
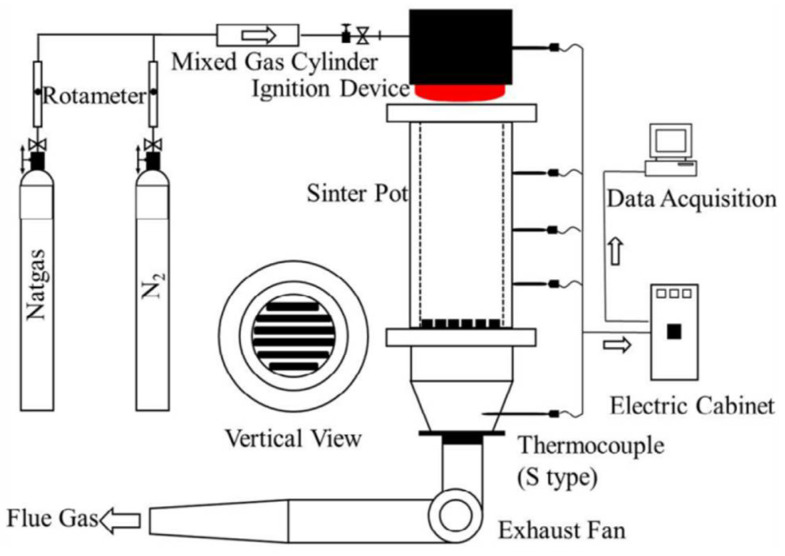
Schematic of sintering pot test equipment.

**Figure 2 materials-14-04376-f002:**
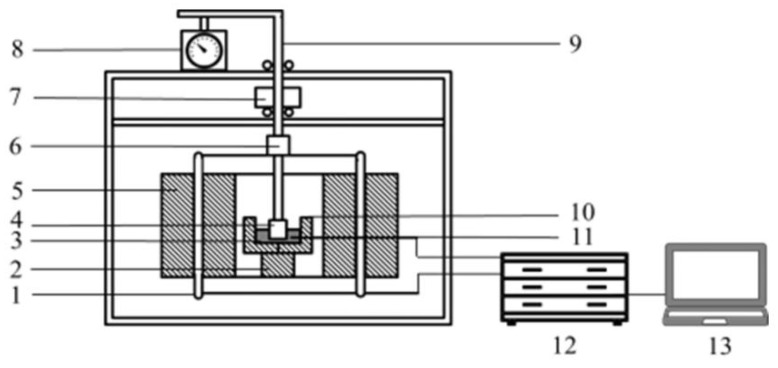
Schematic diagram of the softening properties experimental apparatus [9]. 1—Si-C heater; 2—shell; 3—Al_2_O_3_ pedestal; 4—sample; 5-Si—C bar; 6—fastening screws; 7—load; 8—m; 9—steel bar; 10—bracket; 11—graphite crucible; 12—thermocouple; 13—temperature program controller.

**Figure 3 materials-14-04376-f003:**
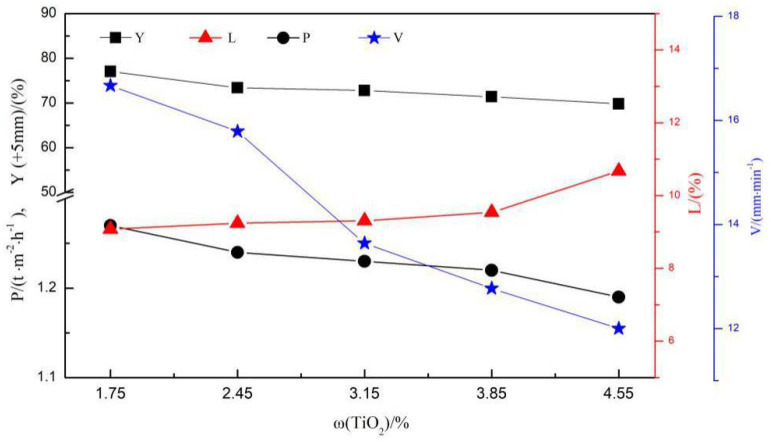
Sintering process parameters of the LVTM sinter.

**Figure 4 materials-14-04376-f004:**
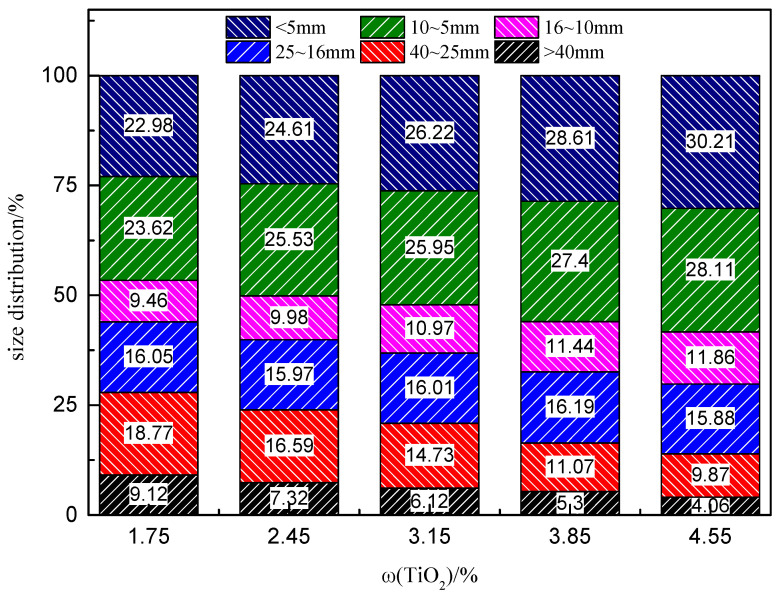
Size distribution of the LVTM sinter.

**Figure 5 materials-14-04376-f005:**
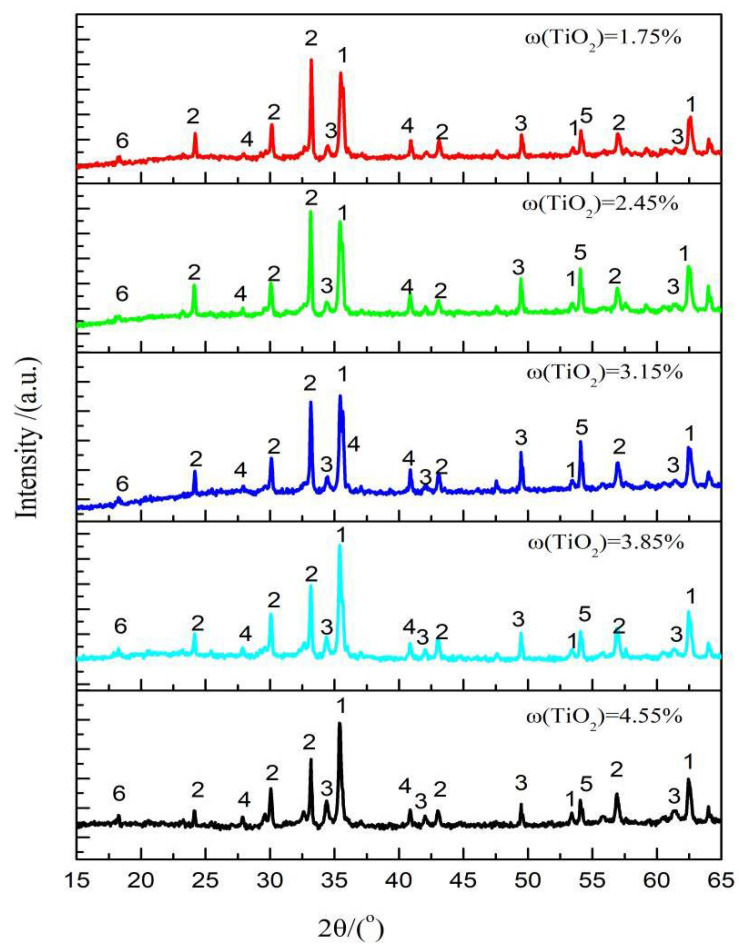
XRD of the LVTM sinter at different TiO_2_ content: 1: magnetite; 2: hematite; 3: calcium ferrite; 4: perovskite; 5: silicate; 6: ilmenite.

**Figure 6 materials-14-04376-f006:**
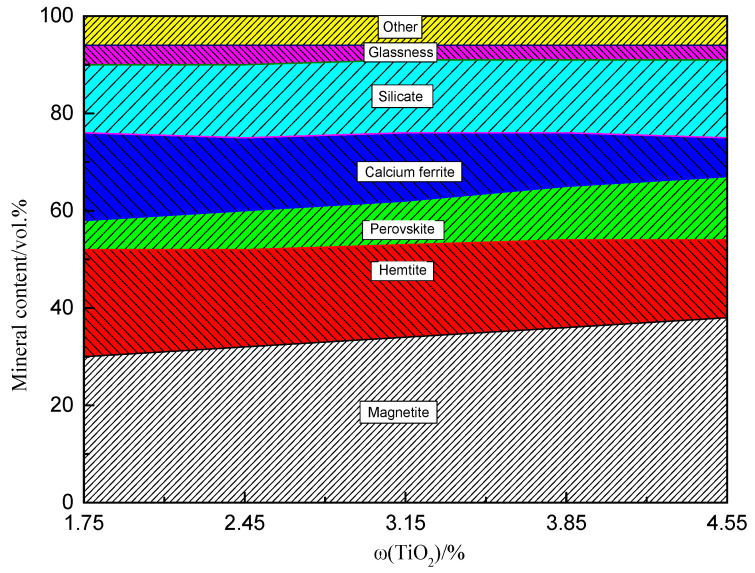
Mineral composition of the LVTM sinter with different TiO_2_ contents.

**Figure 7 materials-14-04376-f007:**
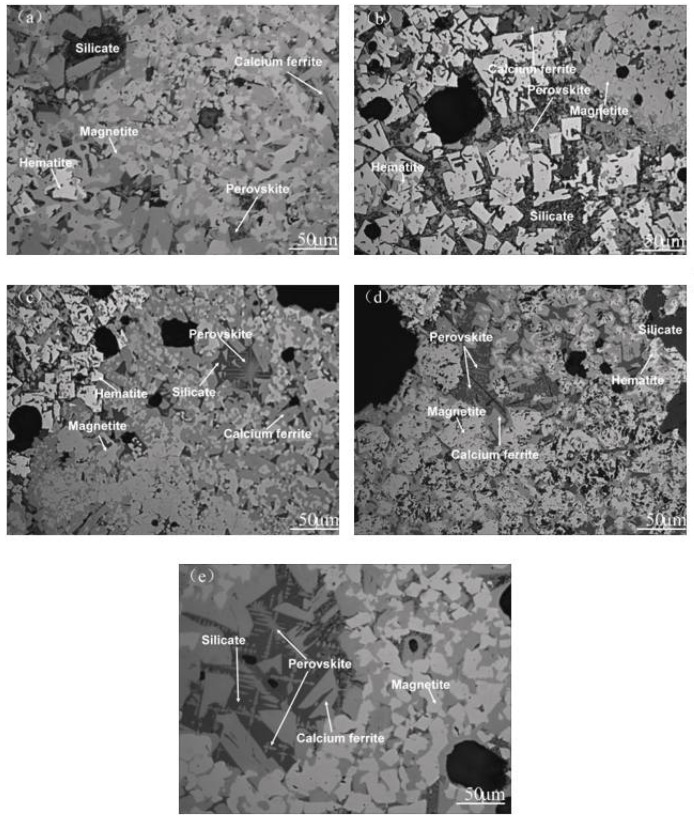
Microstructure of the sinter with various TiO_2_ contents: (**a**) *w* (TiO_2_) = 1.75%, (**b**) *w* (TiO_2_) = 2.45%, (**c**) *w* (TiO_2_) = 3.15%, (**d**) *w* (TiO_2_) = 3.85%, (**e**) *w* (TiO_2_) = 4.55%.

**Figure 8 materials-14-04376-f008:**
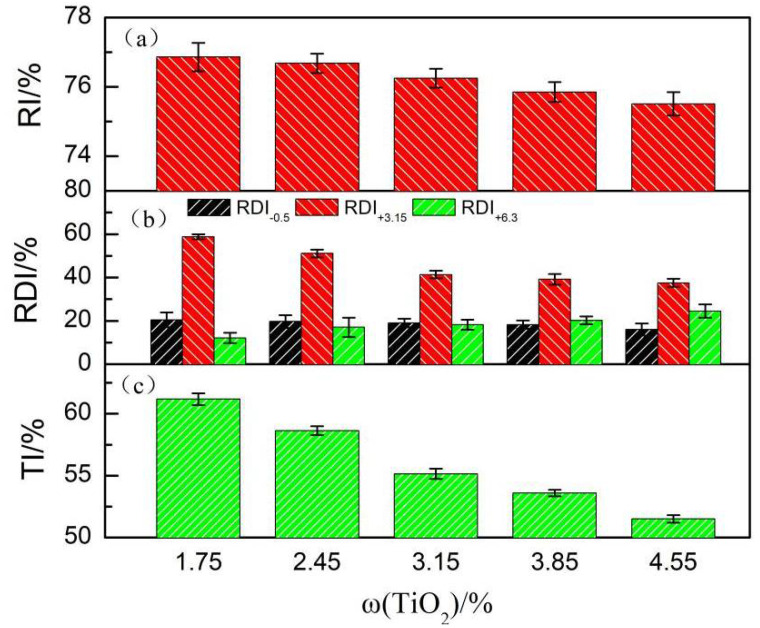
Metallurgical properties of the sinter with various TiO_2_ contents:(**a**) RI, (**b**) RDI, (**c**) TI.

**Figure 9 materials-14-04376-f009:**
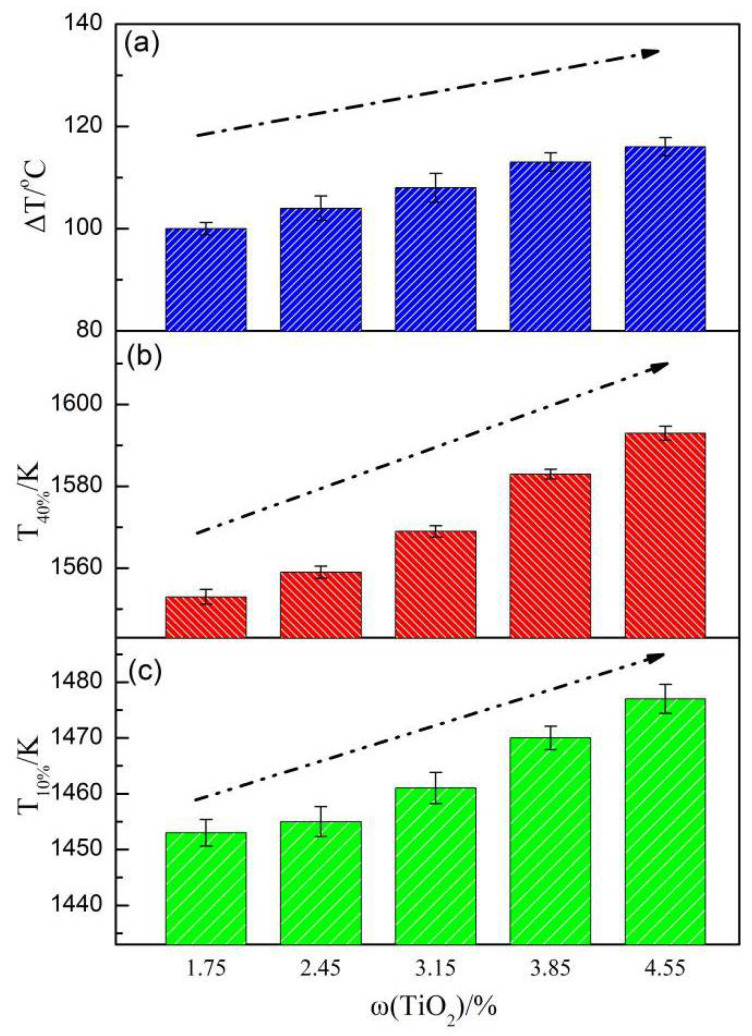
Softening index of the LVTM sinter with various TiO_2_ contents: (**a**) ΔT, (**b**) T_40%_, (**c**) T_10%_.

**Figure 10 materials-14-04376-f010:**
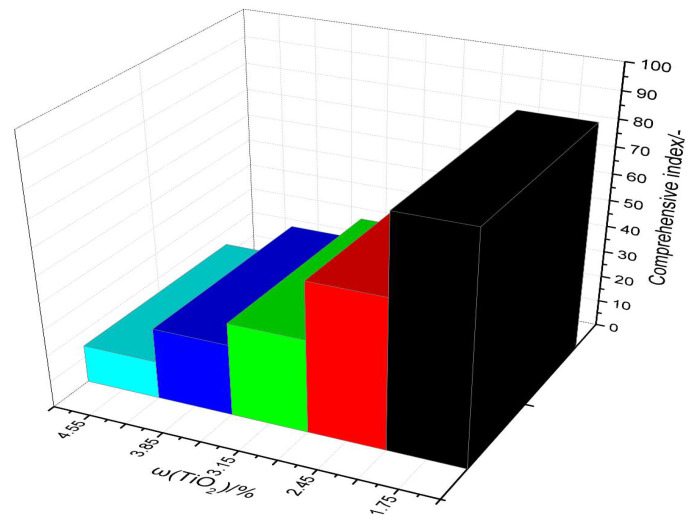
Comprehensive evaluation results of the sinter with different TiO_2_ contents.

**Table 1 materials-14-04376-t001:** Chemical compositions of the raw materials (wt, %).

Item	TFe	SiO_2_	CaO	MgO	Al_2_O_3_	TiO_2_	V_2_O_5_	Cr_2_O_3_
LVTM	63.50	3.96	1.46	1.25	1.57	2.18	0.50	0.08
HT ore	59.61	4.11	0.87	1.68	2.26	6.22	0.70	-
Indian ore	56.06	5.57	0.06	0.15	5.63	0.21	-	-
Malaysian ore	51.71	6.57	0.21	0.15	8.48	0.33	-	-
Zirong ore	65.55	3.04	0.46	3.50	0.65	0.09	-	-
Gas ash	33.28	7.26	5.65	1.98	4.55	1.32	0.25	-
V tailings	30.68	16.97	2.44	2.82	1.53	9.81	1.22	-
Dolomite	-	2.38	45.14	32.05	-	-	-	-
Quicklime	-	2.49	83.17	3.48	-	-	-	-

**Table 2 materials-14-04376-t002:** Proximate analysis of the coke breeze and the ash composition (wt, %) [8].

Fixed Carbon	Total Sulfur	Volatile	Ash (14.00)						∑
			FeO	Al_2_O_3_	SiO_2_	CaO	MgO	Others	
84.00	0.50	1.50	0.14	2.72	7.50	0.48	0.15	2.89	100.00

**Table 3 materials-14-04376-t003:** Parameters of the sintering test.

Item	Parameter	Item	Parameter
Sintering pot size	Φ320 mm × 700 mm	Ignition suction	6.0 kPa
Sintering suction	10.0 kPa	Granulation time	10 min
Ignition temperature	1373 K	Ignition time	1200 S
Moisture	7.5 ± 0.3%		

**Table 4 materials-14-04376-t004:** Experimental scheme of the sinter materials.

Item	*w* (TiO_2_)/%	Mixed Sinter Materials/(100%)								
		LVTM	HT Ore	Indian Ore	Malaysian Ore	Zirong Ore	Return Fine	Ash	V Tailings	Dolomite
1	1.75	64.0	0	5	4	5	18	1	2	1
2	2.45	48.7	15.3	5	4	5	18	1	2	1
3	3.15	33.7	30.3	5	4	5	18	1	2	1
4	3.85	18.8	45.2	5	4	5	18	1	2	1
5	4.55	4.0	60	5	4	5	18	1	2	1

**Table 5 materials-14-04376-t005:** Temperature and atmosphere of the softening properties test.

Item	Parameter		
Temperature	Room temperature–1173 K	1173–1293 K	1293–1823 K
Time	87.5 min	40 min	106 min
Atmosphere	3 L/min N_2_	9 L/min N_2_, 3.9 L/min CO, and 2.1 L/min CO_2_	10.5 L/min N_2_ and 4.5 L/min CO

**Table 6 materials-14-04376-t006:** Chemical composition of the sinter (wt, %).

Items	TiO_2_	TFe	CaO	MgO	SiO_2_	Al_2_O_3_	V_2_O_5_
1	1.75	56.35	7.562	2.08	3.98	1.61	0.32
2	2.45	56.13	7.657	2.09	4.03	1.64	0.36
3	3.15	56.04	7.885	2.11	4.15	1.73	0.39
4	3.85	55.57	8.094	2.13	4.26	2.11	0.44
5	4.45	55.37	8.398	2.19	4.42	2.35	0.51

## Data Availability

Not applicable.

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
