# Peer review of "Effect of TiO2 on the Sintering Behavior of Low-Grade Vanadiferous Titanomagnetite Ore"

_materials, 2021, doi:10.3390/ma14164376_

Round 1

Reviewer 1 Report

The manuscript concerns the processing of vanadiferous titanomagnetite ore. The set of raw materials were mixed in such proportions so that the total concentration of titania varied from 1.75 to 4.55wt.%. The material was then used in the sintering pot test. The manuscript provides technical information about the processing of the ores and the text is logically organized. However, there are some fragments that should be revised.

In the introduction, page 2, lines 47-51 – as I understand, the first part regards the crystallographic system which is followed by the description of the shape of crystals. This part could be modified and provided with clearer description.

Page 3, line 86-87 – the description of the table 4 content is inadequate (the table concerns the materials composition in the sintering pot tests).

Table 6 – it should be mentioned in text that the “items” in the table 6 are in fact the “items” with the composition described in table 4.

Page 5, table 5 – why is the atmosphere changed for various temperature ranges in the softening test? Does the atmosphere influence the softening of the material? Or is it the apparatus operation conditions? Please, supplement the description of the softening experiment with a proper information.

Page 6, lines 139-142 – The information given in this paragraph is quite obvious, as the “items” consisted of different raw materials mixture (with increasing content of the TiO2-rich material). Was the chemical composition calculated or measured (what method was used?). The text should be supplemented with a proper information.

Figure 3 – Please, give the legend (in case of black and white prints)

Page 8, line 168 – please correct “ferrate” to “ferrite”

The conclusions 1 and 3 are in contradiction. In the first conclusion it is stated that the high TiO2 contnent “improves” the parameters, while in the last it is claimed that the low content is more beneficial. The conclusions generally could be more explicit and expanded.

General remarks:

  • it would be helpful for readers to provide explanation to the abbreviations. Usually, the explanations should be repeated at least in every chapter with the first use of the abbreviation. Some abbreviations used in the manuscript lack explanation at all (BF process, CVTM, yield +5mm, TFe – table 1, RDI+3.15).
  • The measured parameters should be better explained and given a short definition.

Author Response

Dear Reviewer:

     Thank you for your letter and for the reviewers’ comments concerning our manuscript entitled “Effect of TiO2 on the Sintering Behavior of Low-grade Vanadiferous Titanomagnetite ore”.

Those comments are all valuable and very helpful for revising and improving our paper, as well as the important guiding significance to our researches. We have studied comments carefully and have made correction which we hope meet with approval. Particularly, I sincerely thank the editors for they provide to me the very professional and careful reviewers in my paper .

But first of all, we will show the modification of this paper and give the description of the modification. The main corrections in the paper and the responds to the reviewer’s comments are as follows:

comments: 

In the introduction, page 2, lines 47-51 – as I understand, the first part regards the crystallographic system which is followed by the description of the shape of crystals. This part could be modified and provided with clearer description.

Response:  Yes, we have reworked this section.“The perovskite was generally cubic or octahedral, with an ideal cubic microstructure and a slightly twisted octahedron. The titanium ions was at the boby center of the cubic cell, the oxygen ions was at the face center and the calcium ions was at the top of the corners. The perovskite cubic crystals often had parallel crystal edges, which was the result of flake bicrystals at  the process of the high-temperature variant changes to the low-temperature variant.”

comments: 

Page 3, line 86-87 – the description of the table 4 content is inadequate (the table concerns the materials composition in the sintering pot tests).

Response:  Yes, we have inserted more descriptions of Table 4 in that paragraph."The mass percentage content of Indian ore, malaysian ore, zirong ore, return fine, ash, V tailings and dolomite for all experimental groups was 5%, 4%, 5%, 18%, 1%, 2% and 1%, respectively, while the mass percentage content of LVTM ranged from 0% to 64% and the mass percentage content of HT ore ranged from 0% to 60%. The TiO2 content of the experimental feedstock was adjusted by adjusting the mass percentage content of LVTM and HT ore.”

comments: 

Table 6 – it should be mentioned in text that the “items” in the table 6 are in fact the “items” with the composition described in table 4.

Response:  Yes. "Table 6 shows the chemical composition of the sinter" was modified to "The composition of sinter with different TiO2 contents was Obtained by Sinter pot experiments according to the experimental scheme in Table 4, as shown in Table 6"

comments: 

Page 5, table 5 – why is the atmosphere changed for various temperature ranges in the softening test? Does the atmosphere influence the softening of the material? Or is it the apparatus operation conditions? Please, supplement the description of the softening experiment with a proper information.

Response: Yes. "Table 5 shows the and atmosphere of the softening properties test" was modified to “According to the actual temperature and atmosphere of sinter in the blast furnace and considering the safety of experimental equipment, the experimental atmosphere and temperature parameters are formulated, as shown in Table 5.”

comments: 

Page 6, lines 139-142 – The information given in this paragraph is quite obvious, as the “items” consisted of different raw materials mixture (with increasing content of the TiO2-rich material). Was the chemical composition calculated or measured (what method was used?). The text should be supplemented with a proper information.

Response: Yes, we have added the methods of analysis of raw materials to this section.“The X-ray fluorescence (XRF, ZSXPrimus II; Rigaku, Tokyo, Japan) was used to test the chemical compositions of raw materials.”

comments: 

Figure 3 – Please, give the legend (in case of black and white prints)

Response: Yes, we have redrawn Figure 3.

comments: 

Page 8, line 168 – please correct “ferrate” to “ferrite”

Response: Yes, we have modified it.

comments: 

The conclusions 1 and 3 are in contradiction. In the first conclusion it is stated that the high TiO2 contnent “improves” the parameters, while in the last it is claimed that the low content is more beneficial. The conclusions generally could be more explicit and expanded.

Response: Yes, we have rewritten the conclusion to make it clearer.

comments: 

General remarks:

  • it would be helpful for readers to provide explanation to the abbreviations. Usually, the explanations should be repeated at least in every chapter with the first use of the abbreviation. Some abbreviations used in the manuscript lack explanation at all (BF process, CVTM, yield +5mm, TFe – table 1, RDI+3.15).
  • The measured paameters should be better explained and given a short definition.

Response: 

Yes, we have fully explained the abbreviations and parameters in the manuscript. Please see the revised draft for details.

We tried our best to improve the manuscript and made some changes in the manuscript. These changes will not influence the content and framework of the paper.

We appreciate for Editors/Reviewers’ warm work earnestly, and hope that the correction will meet with approval.

Once again, thank you very much for your comments and suggestions.

Sincerely

S.T. Yang

Reviewer 2 Report

The article is interesting as it is a contribution to the few studies aimed at defining the possibilities for the future use of low-grade vanidiferous titanomagnetite ore (LVTM), in orderto reduce production costs in the iron and steel industry.

However, the main shortcoming of the work is that its authors write for a very specialised readership and assume as well-known concepts and acronyms, which are not explained in the paper. In order to make the paper interesting to a wider audience, the authors should make an effort to better explain the following concepts:

- Page 2, Line 41: ‘BF process’. The authors should explain what this process consists of.

- Page 3, Section 2.2., Authors should explain the ‘Sintering pot test technique’, its control parameters, etc., otherwise, concepts as ‘flame front’, and the indices defined below (V, L, Y and P) are not easily understood.

In this section 2.2, or where the authors deem appropriate, they should explain what the abbreviations RDI+3.15, RDI, RI and TI used in section 3.4 (without any explanation).

- Page 9, Line 214, Footnote to Fig 5. Authors should clarify what HCVTS stands for.

- Page 10, Line 217, Footnote to Fig 6, Authors should clarify what CCVT stands for.

- Page 11, Lines 224,225,238,240,246 Authors should clarify what RDI+3.15, RDI, RI, TI stand for, if not already done elsewhere above.

- Page 13, Line 266 and 267, Authors should advance an explanation of 'Delphi method' and 'entropy ,method' in section 2.

- Page 13, line 270, Authors should explain what the evaluation matrix, 'the criterion matrix', alpha, beta and other parameters are, what the method consists of, is it a weighted regression adjustment method?

- Page 15, Line 293. The authors should quantify the variations of each of the parameters involved.

- The conclusions are too predictable. I recommend that they add a few more, which would do justice to the effort made.

Some minor errors:

- Line 78. There is an extra . after [8, 9].

- Page 14, Figure 10. The title of the vertical axis appears truncated.

Author Response

Dear Reviewer:

     Thank you for your letter and for the reviewers’ comments concerning our manuscript entitled “Effect of TiO2 on the Sintering Behavior of Low-grade Vanadiferous Titanomagnetite ore”.

Those comments are all valuable and very helpful for revising and improving our paper, as well as the important guiding significance to our researches. We have studied comments carefully and have made correction which we hope meet with approval. Particularly, I sincerely thank the editors for they provide to me the very professional and careful reviewers in my paper .

But first of all, we will show the modification of this paper and give the description of the modification. The main corrections in the paper and the responds to the reviewer’s comments are as follows:

comments: 

- Page 2, Line 41: ‘BF process’. The authors should explain what this process consists of.

Response: Yes, we have explained the BF process.

comments: 

- Page 3, Section 2.2., Authors should explain the ‘Sintering pot test technique’, its control parameters, etc., otherwise, concepts as ‘flame front’, and the indices defined below (V, L, Y and P) are not easily understood.

In this section 2.2, or where the authors deem appropriate, they should explain what the abbreviations RDI+3.15, RDI, RI and TI used in section 3.4 (without any explanation).

Response:Yes, we have reworked and explained the experimental details of the sintering tank in section 2.2, which will help the reader to understand that section.

comments: 

- Page 9, Line 214, Footnote to Fig 5. Authors should clarify what HCVTS stands for.

- Page 10, Line 217, Footnote to Fig 6, Authors should clarify what CCVT stands for.

Response: We apologise that this was a misspelling and we have corrected it in the revised version.

comments: 

- Page 11, Lines 224,225,238,240,246 Authors should clarify what RDI+3.15, RDI, RI, TI stand for, if not already done elsewhere above.

Response: We have re-explained the terms RDI+3.15, RDI, RI, TI in section 2.2 to enable the reader to better understand the manuscript.

comments: 

- Page 13, Line 266 and 267, Authors should advance an explanation of 'Delphi method' and 'entropy ,method' in section 2.

- Page 13, line 270, Authors should explain what the evaluation matrix, 'the criterion matrix', alpha, beta and other parameters are, what the method consists of, is it a weighted regression adjustment method?

Response: We have added some explanations so that the reader can better understand the manuscript.

comments: 

- Page 15, Line 293. The authors should quantify the variations of each of the parameters involved.

Response: Yes, we have added some content.As the TiO2 content of the sinter gradually increases from 1.75 to 4.55, the comprehensive index was 85.63, 56.71, 35.43, 27.17, 14.37 respectively.

comments: 

- The conclusions are too predictable. I recommend that they add a few more, which would do justice to the effort made.

Response: Yes, we have rewritten the conclusion to make it clearer.

comments: 

- Line 78. There is an extra . after [8, 9].

- Page 14, Figure 10. The title of the vertical axis appears truncated.

Response: Yes, we have made changes.

We tried our best to improve the manuscript and made some changes in the manuscript. These changes will not influence the content and framework of the paper.

We appreciate for Editors/Reviewers’ warm work earnestly, and hope that the correction will meet with approval.

Once again, thank you very much for your comments and suggestions.

Sincerely

S.T. Yang